# Designing Nanomedicines for Breast Cancer Therapy

**DOI:** 10.3390/biom13101559

**Published:** 2023-10-22

**Authors:** Saba Abbasi Dezfouli, Amarnath P. Rajendran, Jillian Claerhout, Hasan Uludag

**Affiliations:** 1Faculty of Pharmacy and Pharmaceutical Sciences, University of Alberta, Edmonton, AB T6G 2V2, Canada; sabbasid@ualberta.ca; 2Department of Chemical and Materials Engineering, Faculty of Engineering, University of Alberta, Edmonton, AB T6G 2V2, Canada; amarnath@ualberta.ca; 3Department of Biological Sciences, Faculty of Science, University of Alberta, Edmonton, AB T6G 2V2, Canada; jclaerho@ualberta.ca; 4Department of Biomedical Engineering, Faculty of Medicine and Dentistry, University of Alberta, Edmonton, AB T6G 2V2, Canada

**Keywords:** breast neoplasm, siRNA delivery, RNA therapeutics, gene silencing, polyethyleneimine, survivin, BIRC5, lipophilic polymers

## Abstract

In 2020, breast cancer became the most diagnosed cancer worldwide. Conventional chemotherapies have major side effects due to their non-specific activities. Alternatively, short interfering RNA(siRNA)-carrying nanoparticles (NPs) have a high potential to overcome this non-specificity. Lipid-substituted polyethyleneimine (PEI) polymers (lipopolymers) have been reported as efficient non-viral carriers of siRNA. This study aims to engineer novel siRNA/lipopolymer nanocomplexes by incorporating anionic additives to obtain gene silencing through siRNA activity with minimal nonspecific toxicity. We first optimized our polyplexes in GFP+ MDA-MB-231 cells to effectively silence the GFP gene. Inclusion of phosphate buffer with pH 8.0 as complex preparation media and N-Lauroylsarcosine Sodium Salt as additive, achieved ~80% silencing with the least amount of undesired cytotoxicity, which was persistent for at least 6 days. The survivin gene was then selected as a target in MDA-MB-231 cells since there is no strong drug (i.e., small organic molecule) for inhibition of its oncogenic activity. The qRT-PCR, flow cytometry analysis and MTT assay revealed >80% silencing, ~95% cell uptake and >70% cell killing by the same formulation. We conclude that our lipopolymer can be further investigated as a lead non-viral carrier for breast cancer gene therapy.

## 1. Introduction

Cancer has been defined by World Health Organization (WHO) as the uncontrolled growth of abnormal cells [1]. Breast cancer surpassed lung cancer as the most commonly diagnosed cancer worldwide in 2020, and it is now the second leading cause of death among females worldwide [2]. Conventional therapies like surgery, radiotherapy, chemotherapy, endocrine therapy (hormone therapy) and targeted therapy are available for breast cancer. They, however, have major short-term and long-term side effects such as skin problems (e.g., soreness, itching, peeling) in the area of treatment or reduced sensation in breast tissue in the case of radiotherapy or gastrointestinal disturbance, hair loss and depressed immunity in the case of chemotherapy [3], which are mainly due to their inability to specifically target malignant cells. This is why novel treatment approaches are being explored [4]. The advent of nanotechnology has been foreseen as a potential candidate for cancer therapy since it can provide confined cytotoxicity to cancerous cells resulting in the targeted destruction of malignant tissues and reduced arbitrary drug distribution [5]. NPs benefit from many favorable characteristics such as lower elimination rate, increased drug-site contact time and decreased drug resistance [3]. The NP drug carriers, also known as vectors, comprise two main components: the active drug and the material that forms the NP itself and can be used to improve systemic availability [6,7,8]. Through the ease of conjugation of various helping moieties via a linker, NPs provide a framework for tailoring custom therapy designs. Furthermore, they enable the development of a multifunctional platform capable of incorporating multiple therapeutic agents for simultaneous delivery and effective treatment. On the flip side, their small size can cause adverse effects by allowing them to penetrate biological structures, disrupt their normal activity and cause harmful effects such as tissue inflammation and a shift in cellular redox balance toward oxidation, resulting in irregular functions, immune stimulation or cell death. Yet, when it comes to encapsulating various kinds of drugs, NPs represent incredibly versatile tools [3].

The NPs can be utilized to deliver short-interfering ribonucleic acids (siRNAs). In this strategy, a synthetic double-stranded siRNA is delivered to implement the naturally occurring RNA interference (RNAi) mechanism to silence or downregulate the expression of a desired protein [9]. Outside the cell and in circulation, the NPs bearing the siRNA should stay intact. After being delivered to the cell, though, the siRNA needs to be freed from the endosome, then it will incorporate into the RNA-induced silencing complex (RISC) before its passenger strand is removed and the remaining “guide” siRNA strand directs the RISC assembly to targeted mRNA, followed by either cleavage or blockage of the mRNA to inhibit protein translation [10,11]. However, due to the fragile (degradable) nature of anionic siRNA in physiological conditions, the efficacy of siRNA delivery is heavily dependent on its carrier to successfully transport it to the cell and protect it against degradation by nucleases [12]. Structural motifs, chemical medications and sequence selectivity have been used to design more effective RNAi drugs. Different motifs may have very different functions that will affect the RNAi processing efficiency [13]. For instance, it has been shown that in the case of asymmetric siRNA, the bias for guide strand selection to enter the RISC is toward the strand with a 3′ overhang [14]. Chemical modifications, excluding ligand targeting, have two primary purposes: improved safety by attenuating activation of immune sensors, and increased potency by enhancing protection against degradation by nucleases. Sugar, base, and backbone modifications can be used in this regard [13]. As an example, extensive 2′-O- methyl base modifications on recent RNAi triggers have largely avoided immunogenic reactions that usually result from sensing the double-stranded RNA (dsRNA) by PKR, Toll-like receptor 3 (TLR3) and TLR7 [15]. Finally, sequence selectivity has played an important role in selecting the right strand, the antisense strand, as the guide strand for incorporation into the RISC [13]. It has been demonstrated that a strand with weaker base-pairing as its 5′-end will be chosen to enter the RISC [14,16]. Both viral and non-viral vectors can also be used to deliver siRNA to the cells. Although viral vectors like adenoviruses, lentiviruses and retroviruses, have high transduction efficiency, they are worrying considering their potential for insertional mutagenesis and unwanted immunogenicity. Non-viral vectors, on the other hand, are preferred by virtue of their easy-to-engineer nature and lower manufacturing costs. Non-viral vectors, additionally, can solve other limitations of their viral counterparts by having unlimited loading capacity and providing compatible transient gene expression [17]. Nonetheless, poor transfection efficiency and significant cytotoxicity are still limiting their therapeutic use [18].

Lipidic (e.g., liposomes and niosomes) and polymeric carriers have been explored as non-viral carriers [17]. Cationic polymers have been extensively investigated due to their unique physio-chemical properties that can electrostatically form complexes with nucleic acids. They can protect against enzymatic degradation, mediate transfection via nucleic acid condensation and facilitate cellular uptake and endosomal release. Polyethyleneimine (PEI) is one of the most prominent synthetic cationic polymers with primary, secondary, and tertiary amino functions which is synthesized in linear and branched forms in different molecular weights [19], PEI has repeatedly demonstrated high transfection efficiency [20] which is attributed by its ability to facilitate endosomal escape through the so-called “proton sponge” phenomenon (i.e., uncharged Ns acting as sponges during the endosome maturation that leads to swelling of the polymer, Cl- counterion flux into the endosome and endosome destabilization due to osmotic forces) [21]. The excess positive charges in high molecular weight PEI (>20 kDa) induce plasma membrane destruction which causes loss of metabolic activity and cell death. The low molecular weight PEI (<2.0 kDa) is reported to be less toxic and thus is a more suitable carrier for siRNA delivery. We have previously shown that lipidic substitutions on the PEI backbone can improve the uptake of siRNA/PEI complexes, presumably due to their enhanced chemical compatibility with the cell membrane [22,23].

In this study, we explored leading PEI-Lipid conjugates as polymeric non-viral vectors that include the low molecular PEI (1.2 kDa) grafted with three different lipids: linoleic acid (3-18-C), Oleic acid (1-18-C) and Lauric acid (12-C) via gallic acid (GA) as linker, which allows for three lipid conjugates at a single site of PEI [24]. We inspected their efficacy for siRNA delivery in breast cancer treatment. Our goals were to achieve high gene silencing via siRNA activity while minimizing nonspecific toxicity through the development of effective lipid-modified PEI carriers (lipopolymers). To address these, we hypothesize that lowering the lipopolymer ratio and adding negatively charged additives to our nano complexes will, respectively, reduce cytotoxicity and improve gene silencing ability by facilitating cellular uptake. To scrutinize our hypothesis, we compared the lipopolymer/siRNA with lipopolymer/siRNA/additives for which we selected a range of additives and investigated their efficiency in various polymer:additive:siRNA formulations. Finally, the effect of different buffers for complex preparation was also explored to provide a better siRNA delivery environment.

## 2. Materials and Methods

### 2.1. Materials

Branched 1.2 kDa PEI (bPEI1.2) was obtained from Polyscience, Inc (Warrington, PA, USA). Methylthiazolyldiphenyl tetrazolium bromide (MTT), Dulbecco’s Phosphate Buffered Saline (PBS), Linoleoyl chloride, Lauroyl chloride and Poly(acrylic acid) (PA) [MW: ~2000 Da] were obtained from Sigma-Aldrich (St. Louis, MO, USA). Green Fluorescent Protein (GFP) positive MDA-MB-231 (MDA-MB-231-GFP+) cell line was prepared through retroviral transfection [25]. Dulbecco’s Modified Eagle Medium F12 (DMEM F12), dimethyl sulfoxide (DMSO) was obtained from MiliporeSigma (Saint Louis, MO, USA). GFP-siRNA, negative control scrambled siRNA, 6-‘(FAM)-labeled scrambled siRNA, and survivin siRNA were obtained from Integrated DNA Technologies Inc. (Coralville, IA, USA) and their sequences are mentioned in Appendix A Table A1. N-2-hydroxyethylpiperazine-N-2-ethane sulfonic acid (HEPES) (BP 310-500) was purchased from Fisher Bioreagents (Pittsburgh, PA, USA). Trypsin was purchased from ThermoFisher Scientific Inc. (Waltham, MA, USA). Other additives including β-Glycerophosphate Disodium Salt Hydrate (GDS), Potassium Phosphate Monobasic (PPM0, Sodium Pyrophosphate (SPP), tri- Sodium Orthophosphate (TSO) and N- Lauroylsarcosine Sodium Salt (LS), as well as the 1/15 mole L^−1^ phosphate buffer, 0.1 mol L^−1^ Citrate buffer, and PEI-LA transfection reagent were prepared in-house. To create 3.7% formaldehyde, Hank’s Balanced Salt Solution (HBSS) was used to dilute a 37% stock solution, which was purchased from Sigma-Aldrich.

### 2.2. Cell Culture

The MDA-MB-231 breast cancer cells were cultured in DMEM/F12 medium containing 10% fetal bovine serum (FBS) as well as 100 U/mL of penicillin, and 100 μg/mL of streptomycin and 5 mL sodium pyruvate. The cells were maintained at 37 °C under humidified conditions with 95/5% air/CO_2_. The cells were routinely examined for mycoplasma contamination using a PCR-based method as explained by Young et al. [26] and validated by STR DNA profiling analysis at the Genetic Analysis Facility, The Centre for Applied Genomics, The Hospital for Sick Children (Toronto, ON, Canada). At around 80% confluency, cells were seeded 24 h before treatment with siRNA complexes as follows. After removing the utilized media and rinsing with 5 mL PBS, 1 mL of 0.05% trypsin was added and the cells were incubated at 37 °C for 2–3 min to promote cell dissociation and were diluted by the addition of 5 mL DMEM. Subsequently, the cells were centrifuged at 600 rpm for 5 min, followed by resuspension in 5 mL fresh DMEM. Finally, 300 µL of cells were seeded in 48-well plates at the density of 60,000 cells/mL for GFP gene experiments and 20,000 cells/mL for survivin gene experiments.

### 2.3. siRNA Complex Preparation

Citrate buffer (0.1 M) was prepared by mixing disodium citrate (0.1 M; 1 L solution contains 21.01 g citric acid monohydrate + 200 mL NaOH 1 M) and HCL 0.1 M. Phosphate buffer was prepared by mixing potassium dihydrogen phosphate 1/15 M (9.073 g L^−1^) with disodium phosphate 1/15 M (11.87 g/L). All the buffers were filtered with a syringe before use. Additives (0.14 ug/uL) were added to the media after siRNA (0.14 µg/µL) had been dissolved in the media and before the transfection reagent (PEI-LA 1 mg/mL or PEI-GA-Lau7 0.5 µg/µL) was added at desired (*w*/*w*/*w*) ratio of transfection reagent to additive to siRNA. The PEI-LA and PEI-GA-Lau7 lipopolymers were prepared as described in our earlier papers. (Figure 1) [24,27]. Ca^2+^ (0.14 µg/µL) was then added in the form of CaCl_2_ if required with the ratio 1:1 to siRNA. The solutions were then incubated at room temperature for 30 min for optimal complexation. Finally, the remaining amount of buffer was added to make 200 µL of the solution with the final siRNA concentration of 50 nM in tissue culture medium. The specific concentrations used are indicated in Figure legends. Table 1 shows a sample design for the siRNA complex preparation.

### 2.4. Physiochemical Characterization of siRNA Polyplexes

The particle size and surface charge (𝜁-potential) of siRNA polyplexes were determined by Litesizer 500 (Anton-Paar). The complexes were prepared in 200 µL as described above and were diluted in 1 mL of their corresponding complexation media before measurement. For heparin dissociation experiments, the heparin sodium salt was added to complexes in concentrations ranging from 0 to 10 U/mL for 1 h incubation with the complexes. The complexes were then added to 96-well plates that already had an equal volume of 2X SYBR Green to reach a total volume of 500 uL. The amount of fluorescence was measured with Fluoreskan Ascent 2.5 (Thermo Labsystems) at excitation/emission 485/527 nm. SYBR Green I was used to quantify pure siRNA. Pure siRNA and siRNA+heparin were used as negative and positive controls, respectively.

### 2.5. GFP Silencing

The amount of fluorescence in cells was measured as an indication of GFP gene expression. For this, the cells were rinsed with PBS (1X) after the removal of the media, trypsinized, fixed in 3.7% formaldehyde and then transferred to black 96-well black plates for fluorescence measurement by Fluoroskan Ascent 2.5 (Thermo Labsystems) at excitation 485 nm/emission 527 nm.

### 2.6. Flow Cytometry

To determine the delivery efficiency of siRNA complexes, MDA-MB-231 cells were transfected with FAM-labeled scrambled siRNA at the same concentration either 24 or 48 h after seeding. On day 3 post seeding, the media was removed, and cells were washed with PBS. Then, they were detached and fixed in 3.7% formaldehyde. The no-treatment samples were used as the negative control. The FAM-labeled siRNA positive population and mean fluorescence were quantified using BD Accuri C6 Plus flow cytometer using FL2 channel (10,000 events/sample). The FAM-labeled siRNA-positive population was set as 1% in this case.

### 2.7. RNA Extraction and Quantitative Reverse Transcription PCR (qRT-PCR)

For the qRT-PCR experiment, to quantify the amount of silencing achieved by desired siRNA, MDA-MB-231 cells were first seeded 24 h prior to survivin siRNA transfection. Next, 72 h later, total RNA was extracted by the addition of TRIzol reagent (Invitrogen, Carlsbad, CA, USA) and transferred to cDNA using SensiFAST cDNA Synthesis Kit (Meridian Biosciences Inc., Cincinnati, OH, USA) according to the manufacturer’s instruction. For amplification, 3 µL of cDNA was added to 7 µL of the master mix which itself includes 5 µL of SensiFAST SYBR Hi-ROX reagent (Meridian Biosciences) and 1 µL of each of the forward and reverse primers for either survivin gene or β-actin as the housekeeping gene (Integrated DNA Technologies Inc., Coralville, IA, USA). The sequences for the mentioned primers are as follows: β-actin primers and survivin primers (Appendix A, Table A2). The samples were amplified by StepOne Real-Time PCR System (Applied Biosystems, Foster City, CA, USA) in a denaturation stage (95 °C, 20 sec) followed by 40 cycles at 95.0 °C for 3 sec (denaturation) and annealing and elongation at 60 °C for 30 sec. The results were analyzed using 2^−ΔΔCT^ and presented as a relative quantity of no treatment.

### 2.8. MTT Assay

To perform the assay, 100 µL of filtered MTT solution (5 mg/mL) was added to the 48-well plates at 1 mg/mL. The cells were incubated at 37 °C to convert the yellow MTT to purple formazan crystals using NAD(P)H-dependent oxidoreductase enzyme [28]. After 2 h, the media were removed, and the crystals were dissolved in DMSO for 5–10 min. The solutions were then transferred to 96-well plates and their absorbance was measured at 570 nm by the multi-well spectrophotometer.

### 2.9. Statistical Analysis

The data are presented as mean ± s.d. The results were analyzed by homoscedastic one-tailed distribution *t*-test, where the asterisks (*), (**) and (***) represent significantly different groups with *p* < 0.05, *p* < 0.005 and *p* < 0.0005 in figures in comparison with no treatment and circles (°), (°°) and (°°°) indicate the significant silencing of targeted mRNA transcripts by specific siRNA compared to that of control siRNA with the above-mentioned *p* values.

## 3. Results

### 3.1. PEI-GA-Lau7 Can Deliver GFP-siRNA Better Than PEI-LA to MDA-MB-231 Cells

To determine if the newly developed PEI-GA-Lau7 could be advantageous over the previously optimized PEI-LA, both polymers were compared at polymer:siRNA ratios of 5, 7.5 and 10 with/without PA as an additive in the DMEM complexation medium. GFP-positive MDA-MB-231 cells were analyzed 72 h after transfection. The silencing activity was similar (40–43%) all ratios of PEI-LA and the PEI-GA-Lau7 polymer were as effective as PEI-LA, but the latter polymer showed more toxicity under these conditions (>50% cell death; not shown). Citrate (pH levels from 1 to 5) and phosphate (pH levels from 5 to 8) buffers were examined as complexation media. Lower ratios of 1 and 2 for PEI-GA-Lau7 were used to lower its toxicity, whereas previously optimized ratios of 5, 7.5 and 10 for PEI-LA were used. Several additives were tested to PEI-GA-Lau7 and the GFP fluorescence was measured 72 h after treatment (Figure 2). Cells could not survive the treatment with complexes which were formed in citrate (not shown); thus, this buffer was not tested any further. Regarding the phosphate buffers, the silencing with PEI-LA was similar in pH 5.0 and 8.0 buffers, but cytotoxicity appeared to be higher in the latter buffer. The silencing with the PEI-GA-Lau7 was best in the pH 8.0 buffer with no apparent cytotoxicity at the polymer:siRNA ratio of 1:1. At the higher ratio, the cytotoxicity of PEI-GA-Lau7 generally increased. PA was not particularly beneficial in improving the silencing, but other additives appeared to increase the silencing efficiency to some extent (10–20%), with no obvious candidates among the HEPES, GDS, SPP, PPM, TSO, and LS emerging as a clear-cut choice.

### 3.2. Ratio 1 of Additive Is Better Than Higher Ratios in MDA-MB-231 GFP+ Cells

To further optimize the formulation, higher ratios of additives to siRNA (1:1, 2:1 and 3:1) in phosphate-8.0 were tested while the PEI-GA-Lau7:siRNA ratio was kept at 1:1 (Figure 3). When compared to PEI-LA, PEI-GA-Lau7 caused less cytotoxicity than before. Moreover, the polymer:additive:siRNA ratio of 1:3:1 appeared to be toxic in most cases. PA, which performed best when combined with PEI-LA, underperformed when combined with PEI-GA-Lau7. When inspecting the PEI-GA-Lau7 results, on average, the majority of the selected additives had fairly comparable silencing abilities, HEPES (~45%), GDS (~48%), and PPM (~47%), whereas LS demonstrated the highest silencing (~60%) consistent with Figure 2 results. Apart from PA, ratios 1 and 3 of additives showed the highest and lowest GFP silencing activity, respectively.

The impact of calcium in the formulations was also investigated, following the work of Dick et al. [29] which found a beneficial effect of Ca^2+^ on plasmid DNA (pDNA) complexes and transfection efficiency. The Ca^2+^ was added to the complexes in phosphate-8.0 and cells were examined 3 days following the treatment. No noticeable improvement in toxicity resulted from the incorporation of Ca^2+^ and no beneficial effect of Ca^2+^ could be seen on the silencing efficiency (not shown). Incorporation of Ca^2+^ was deemed non-beneficial and not pursued any further in the case of siRNA delivery.

### 3.3. Physiochemical Characteristics of siRNA Polyplexes

To characterize the physical properties of the particles, their size and zeta potential were measured after the complexes were made in phosphate buffer with different pH with or without additives as usual and were then diluted in the same buffer to reach the total volume of 1 mL (Figure 4A). The particle sizes varied between 200 nm and 1.6 µm. However, most of them were between 500 and 750 nm. When no additive was added, the smallest particle (551 nm) and largest particle (644 nm) were seen in pH 7.0 and 5.0, respectively. pH 6.0 and 8.0 gave similar particle sizes, which were ~600 nm. Regarding the additives effect, contrary to complexes in pH 8.0, additives appeared to increase the particle sizes in most of the cases in pH 5.0–7.0. PA, however, led to the smallest particle size in all pH levels. The polydispersity index (PDI) of the complexes is summarized in Appendix A. Figure A1.

The zeta potentials of the complexes were all between −15 and −2 mV. The complexes prepared in phosphate buffer-5.0 had less negative zeta potentials on average (approximately −4), which was followed by pH 6.0 (~9), 8.0 (~11) and 7.0 (~13). In most of the cases, additives led to a lower value of the zeta potentials.

To further elucidate the characteristics of polyplexes, the stability of the polyplexes against heparin displacement was investigated as a function of increasing heparin concentrations (Figure 4B). The polyplexes were formed by ratio 1:1:1 of PEI-GA-Lau7:additive:siRNA. A lower half-maximal dissociation concentration (DC_50_) was observed at higher pH levels for PEI-GA-Lau7 (i.e., DC_50_ of 1.9 U/mL in pH 8.0 and 4.3 U/mL in pH 5.0). Most of the additive formulations behaved similarly; the additives led to more stable complexes against heparin (at all pH values tested), which was indicated through higher DC_50_ compared to complexes without additives.

### 3.4. siRNA-Mediated GFP Silencing Is Persistent for at Least 6 Days in MDA-MB-231 Cells

The activity of siRNA complexes was then explored for a period of 6 days (Figure 5). When the 2 polymers were compared at various time points without the additives, the PEI-GA-Lau7 silencing ability was higher on days 1 (15%), and 3 (50%), but that of PEI-LA was higher on day 6 (70%). However, more overall silencing was accomplished in combination with additives for PEI-GA-Lau7, with 84% being the highest in the case of PEI-GA-Lau7 + LS on day 3. Interestingly, even with the additives, a similar pattern in performance was visible, with PEI-LA performing at its peak on day 6 and PEI-GA-Lau7 performing more or less similarly from day 3 onward.

### 3.5. Optimized Polyplexes Showed ~95% Cell Uptake in MDA-MB-231 Cells

Flow cytometry was then performed to investigate the cellular uptake of different formulations (Figure 6). The cells were seeded and treated either 24 or 48 h later with the phosphate-8.0 formulated FAM-labeled siRNA at ratios 1:1:1, 3:1:1 or 5:1:1 of PEI-LA or PEI-GA-Lau7:siRNA:additive. Most PEI-GA-Lau7 complexes gave >90% siRNA-positive cells and only 2 of the PEI-GA-Lau7 formulated polyplexes had less than 90% cellular uptake (~80% and ~70% for ratio 1 of PEI-GA-Lau7 to siRNA without additive or with PA as the additive, respectively). However, only 2 of the PEI-LA formulated polyplexes had >90% cellular uptake (ratio 5 of PEI-LA to siRNA when combined with either HEPES or LS). With PEI-LA polyplexes, the uptake was generally increased after 48 h, whereas an opposite trend was observed with PEI-GA-Lau7 polyplexes. 

### 3.6. Polyplexes Can Silence Endogenous Genes with High Efficiency and Low Cytotoxicity

To probe the efficacy of our optimized polyplexes to silence an endogenous gene, survivin was chosen as the target. Wild-type MDA-MB-231 cells were treated with 50 nM survivin siRNA complexes in phosphate-8.0 buffer using either PEI-GA-Lau7 or PEI-LA in ratios 1, 3 or 5 with or without PA, HEPES or LS as the additive to treat. Then, the qRT-PCR assay was performed on day 3 post-treatment (Figure 7). Comparing different ratios of PEI-GA-Lau7, the most amount of silencing was observed with the incorporation of HEPES to ratio 3 of PEI-GA-Lau7:siRNA. Yet, when the same formulation was used in ratio 1 of PEI-GA-Lau7:siRNA, >90% silencing was achieved. For LS-formulated polyplexes; however, ratio 1 proved to be the best after ratio 3 and ratio 5 regarding survivin gene silencing. Except for ratio 1 in which LS and HEPES polyplexes similarly showed minimum toxicity (~−20%), in the other two ratios of PEI-GA-Lau7 to siRNA, HEPES was clearly less toxic (~−50% for HEPES vs. ~35% for LS in ratio 3 and ~6% for HEPES vs. ~35% for LS in ratio 5). In terms of PEI-LA versus PEI-GA-Lau7 comparison, in ratio 1, ~10% and ~40% of toxicity and silencing on average were seen with PEI-LA, both of which were improved with PEI-GA-Lau7 (~−5% toxicity and ~80% silencing). Adding HEPES or LS to PEI-LA polyplexes reduced the toxicity from about 10% to almost −10% in ratio 1 of PEI-LA to siRNA.

### 3.7. Polyplexes Could Effectively Inhibit Cell Growth in MDA-MB-231 Cells

The MDA-MB-231 cells were treated with synthesized polyplexes 24 h after seeding. Ratios 1, 3 and 5 of PEI-LA or PEI-GA-Lau7 and ratio 1 of additives to siRNA were employed in phosphate-8.0 buffer. The cell viability was assessed 72 h after treatment (Figure 8). The best cell-killing activity (>90%) was achieved by the incorporation of HEPES into ratio 1 of PEI-GA-Lau7:siRNA. Treatment with ratio 1 of PEI-GA-Lau7 + LS killed ~70% of the targeted cells. More toxicity was observable in higher ratios of PEI-GA-Lau7 in control siRNA samples. PEI-LA was less toxic in higher ratios with polyplexes formulated with control siRNA. The superior specific killing ability of the PEI-GA-Lau7 alone to PEI-LA can also be observed in all the ratios, especially in ratio 1 (~5% for PEI-LA vs. ~98% for PEI-GA-Lau7).

## 4. Discussion

Nucleic-acid-based therapeutics have been investigated as promising approaches for targeted therapies for several cancers including breast cancer [30,31,32]. siRNAs allow the targeting of specific therapeutic markers and are easy to synthesize as a therapeutic. They may also have a short development time allowing them to readily switch siRNAs that can target different biomarkers. However, until better carriers are created to protect and deliver them intact into the cytoplasm, they will not be able to fulfill their perceived potential for therapeutic outcomes [33]. Carrier development is challenging; they ought to be safe and nontoxic and determine if they can be deployed to other nucleic acid therapeutics [27]. The transfection reagents that have been used here are based on nontoxic low molecular weight (1.2 kDa) bPEI that has been modified with hydrophobic moieties and has previously been studied for pDNA delivery [24]. The membrane destabilization has been suggested as the mechanism for both cytotoxicity and cargo delivery [34,35]; hence, the hydrophobic moieties are expected to improve interactions between the cell membrane and polymeric carrier which in turn will result in higher cellular uptake and, unfortunately, more cytotoxicity [27]. To address this issue, in the present study, one of our main interests was to achieve the highest gene silencing while lowering the non-specific cytotoxicity. We decided to look into lower transfection reagent to siRNA ratios for this purpose. The N/P ratio refers to the ratio of nitrogen atoms in PEI to phosphates in nucleic acid. It has been estimated that in physiological pH, 1 in 5 or 6 Ns of bPEI is protonated and these are the only amines that interact with nucleic acids. Since the pKa of each individual nitrogen cannot be determined, the N/P ratio is used to describe the amount of present polymer in polyplexes and along with zeta potential has been demonstrated to greatly impact the complex effectiveness. Higher ratios will have more interactions with the cell membrane due to their excess positive charges and will lead to more cellular uptake [36,37]. On the other hand, they are reported to form more stable complexes which can act as a barrier inside the cells for siRNA release [38]. In this context, after comparing our newly developed polymer PEI-GA-Lau7 with PEI-LA as our leading breast cancer transfection carrier in previously optimized ratios, we initiated our studies with further optimization of complex formulations. The results showed considerably improved nontoxicity (on average 40% in PEI-LA vs. −10% in PEI-GA-Lau7) as well as enhanced silencing ability (60% for with-HEPES or LS formulated PEI- polyplexeS) for PEI-GA-Lau7 complexes formed at ratios 1 and 2 to siRNA. The flow cytometry also showed improved uptake for the PEI-GA-Lau7 polyplexes and is most likely the underlying basis of improved GFP silencing.

Moreover, it has previously been shown that anionic additives in polyplexes can be beneficial not only for pDNA delivery but also for siRNA [27,39]. It has been reported that contrary to the likelihood of the excess anionic charges hampering the interactions between complexes and cell membrane [40], adding PA to PEI-LA increased the cell uptake [39]. Furthermore, more anionic complexes can also benefit from lower cytotoxicity [41] as well as protecting the siRNA from enzymatic degradation [42]. Parmar et al. have also shown that adding hyaluronic acid (HA), poly(acrylic acid) (PA) or dextran sulfate (DS) to polyplexes as additives can result in more robust siRNA release due to lower polyionic complexation and increased siRNA availability inside the cells. They also observed improved retention time for siRNA particles inside the cells thanks to additive incorporation [27]. Although these carriers were able to deliver their cargo efficiently, there is always a demand for better carriers. That is why one of our objectives for this study was to identify new additives for siRNA complexation. In this regard, we first selected 6 potential additives (HEPES, GDS, SPP, PPM, TSO, LS) in addition to PA. After a series of experiments, LS and HEPES repeatedly demonstrated better performance and were thus chosen for further studies. Considering the combination of PEI-GA-Lau7: siRNA ratio and the additives’ functionality, using ratio 1 of PEI-GA-Lau7 could achieve 48% silencing on average while half of the selected additives showed >50% silencing. In head-to-head comparisons in these experiments, PEI-LA showed relatively lower potency for siRNA mediated silencing. Cytotoxicity was also improved considerably after utilizing lower ratios of PEI-GA-Lau7 (see Figure 2). In addition, the ratio of additives was also optimized for these formulations in an experiment in which siRNA polyplexes were formulated with either ratio 1, 2 or 3 of additives to siRNA (Figure 3). Ratio 1 had the most average silencing (~60%) after ratio 2 (~50%) and ratio 3 (~42%). Although none of the complexes were more than 10% toxic, a similar trend was seen as the ratios of additives went up from 1 to 3.

Another objective of our study was to explore various buffers and systematically see if the ion strength will influence the efficacy of siRNA polyplexes. Different solutions such as HEPES [43,44], water [45], PIPES buffer (20 mM PIPES, 150 mM NaCl) [46], HBG buffer (20 mm HEPES, 5% *w/v* glucose, pH 7.4) [47] and Tris-EDTA [48] have been investigated for nucleic acid complexation. Within this framework, we experimented with water, citrate and phosphate buffers and DMEM as the complexation media, with Ca^2+^ as a special additive due to the previously observed benefit of this ion on nucleic acid complexes. Comparing different pH levels of phosphate buffers, pH 7.0 (~13%) and 8.0 (−8%) were the most and least toxic but phosphate-8.0 showed by far the highest silencing activity compared to other phosphate buffers. The most silencing (~60%) was observed with the inclusion of either HEPES or LS as the additive with the ratio 1 of PEI-GA-Lau7 to siRNA in pH 8.0.

The physiochemical characteristics of particles can dramatically affect their efficacy. Therefore, we probed the effects of pH and additives on particle size and zeta potential. The smallest average size of particles was achieved at pH 8.0. Since larger particles cause adverse effects like microinfarctions [49], pH 8.0 was selected for further studies. The fact that PA resulted in the smallest particles in all four pH levels can be explained by PA being the only polymer among all the additives so it can form more interactions with the PEI polymer to better compact the complexes. Additives, on the other hand, increased the size of the particles in most cases except at pH 8.0. To elaborate, in pH 8.0, the no-additive complex was ~619 nm while the particles with PA, HEPES and LS were ~402, ~412 and ~505 nm, respectively. The particles with other additives were larger.

Zeta potentials were all negative which can be beneficial in the sense that they may cause less toxicity and irreversible damage to the cell [50]. This can, at the same time, decrease the cellular uptake given the negative charge of the cell membrane. However, considering the flow cytometry results, it seems that the anionic charge did not impede the cellular uptake for these complexes. pH 5.0 showed the average zeta potential of ~−4 mV which increased to ~9 mV in pH 6.0, ~−13 mV in pH 7.0 and ~−11 in pH 8.0. This might be because the more acidic the environment is, the more positive charges it has that can push the negatively charged complexes toward the positive side. In pH 8.0, however, the additives show an effect with LS (~−8 mV) pushing the zeta potential to the positive side more than the rest of the additives. PA with ~−15 mV, on the other hand, showed the most negative charges. These patterns are consistent with previous studies [51].

The lower DC_50_ in pH 8.0 can be explained by the abundance of negative charges in more alkaline pH, which will loosen the siRNA-lipopolymer binding and make it easier for heparin to dissociate it. The fact that additives increased DC_50_ was a reflection of additives making more stable complexes in the sense that they could shield the complexes and make it more difficult for heparin to access the polymer. From the dissociation study, it was shown that the PEI-GA-Lau7 complexes in pH 8.0 have a tendency for good dissociation in lower levels of heparin (DC_50_), compared to other pH levels so it can be a better candidate for gene delivery.

We then explored survivin as a target to determine if the obtained results with a reporter gene could be reproduced with the endogenous biomarker. Apoptosis is a cell death programmed with distinct biochemical and morphological characteristics that have been conserved throughout evolution [52]. Impairments in apoptosis lead to many diseases including cancer [53]. It may also adversely impact cancer cells’ response to chemotherapy and radiation, contributing to treatment resistance [54]. Apoptosis induction, whether by stimulating apoptotic pathways or inhibiting antiapoptotic pathways, is being investigated as a cancer therapy strategy. Survivin has been found to be involved in a variety of intracellular mechanisms, all of which promote cell survival [55]. Survivin belongs to the inhibitor of apoptosis proteins (IAP) family, which prevents apoptosis by inhibiting caspase activation [56]. It is the smallest IAP and was first discovered in 1997 [57,58]. Its gene of 14.7 kb which spans on the chromosome 17 telomeric part and is localized to q25 band [57], encodes a 16.5 kDa protein consisting of 142 amino acids. Unlike other mammalian IAP members that usually have two or three Repeats of Baculovirus IAP Repeat (BIR) domain, an essential Cys/His-based zinc finger for apoptosis inhibition, the survivin protein has only one copy of the BIR domain [59]. Contrary to embryonic and fetal organs with strong expression of survivin, in most of the differentiated normal tissues [60], survivin is undetectable, which makes it a perfect target to reduce cell growth solely in malignant cells without affecting normal cells [27]. Overexpression of survivin has been observed in many cancer models and its nuclear expression is associated with its cell division role via controlling the stability of microtubules of the normal mitotic spindle [61]. Survivin has previously been reported as a viable target for breast cancer treatment [62]. Shepherdin [63], YM155 [64], and terameprocol [65,66] are examples of small molecular weight antagonists for survivin that have been investigated for cancer therapy. Nevertheless, due to their poor selectivity as small organic entities, these agents have the potential to have undesirable effects [38]. Several studies have successfully induced apoptosis via survivin knockdown using various RNAi techniques [67,68,69,70]. qRT-PCR revealed >90% silencing in wild-type MDA-MB-231 cells by the inclusion of HEPES and LS in ratio 1 of PEI-GA-Lau7 to siRNA. Whereas LS could only silence ~50% of survivin transcripts in higher ratios. The previously seen low cytotoxicity was also evident in this experiment with ~−3% for PEI-GA-Lau7 and ~10% for PEI-LA on average in ratio 1 to siRNA. Further confirmation of survivin silencing was based on the cell viability of MDA-MB-231 cells after survivin targeting. PEI-LA at its best formulation, PEI-LA/siRNA5.0-HEPES1.0, could achieve ~78% cell death while its other formulations with PEI-LA showed <40% cell killing activity. PEI-GA-Lau7, on the other hand, showed ~98% cell death on its own and ~99% and ~71% when combined with HEPES or LS in the ratio 1 to siRNA (optimal formulationS). Responses in non-malignant breast cells have been compared to cancerous cells in our previous paper. Significant cell growth inhibition by similar formulations in breast cancer MDA-MB-231 cells was reported but not in the MCF-10A cells as the non-malignant breast cells line. Although the non-malignant MCF-10A cells were in fact transfected by the complexes due to morphological similarities, the siRNA uptake was lower compared to malignant cells [27].

It is also noteworthy that a concentration of 50 nM of siRNA was used in all the experiments which is in the concentration range for therapeutic silencing [71,72]. The cell line, carrier and other factors can change this range due to changes in siRNA bioavailability, expected treatment effects and other reasons. Different concentrations of siRNA are being used. Persengiev et al. reported a concentration-dependent specificity for siRNA in HeLa cells. In their study, by using expression profiling, using a range of 0 to 200 nM of luciferase siRNA, they reported 100 nM as the threshold where non-specific effects happen, which can be crucial when the treatments will not lead to cell killing. Lowering the siRNA concentration, however, reduced the silencing efficiency. They finally observed the best efficiency between 25 and 50 nM. Their results indicate the importance of optimizing the siRNA concentration in a way to achieve efficient silencing with the lowest siRNA concentration [72]. Finally, PEI1.2 KDa was excluded as a control in these studies because historically it has been tested in the same cell line in our lab and has shown no transfection efficiency [73].

## 5. Conclusions

In conclusion, we report a robust siRNA delivery system, PEI-GA-Lau7, which can effectively downregulate the expression of the survivin gene in MDA-MB-231 cells. This was accomplished by the incorporation of two newly identified anionic additives, HEPES and LS, into the polyplexes which enhanced the siRNA uptake by the cells. The cytotoxicity was decreased to a minimum by lowering the ratio of PEI-GA-Lau7 to siRNA and the complexation media was also optimized by choosing a phosphate buffer with a pH level of 8.0. Our polymer has proven to be a promising non-viral vector for gene therapy in breast cancer. Nevertheless, similar studies should be conducted in animal models to further validate the beneficial effect of formulated polyplexes. Since ‘stealth’ features might be critical in impacting blood pharmacokinetics favourably and achieving better disease tissue targeting through improved circulation, further engineering of our nanoparticles might be needed for animal studies [74].

## 6. Patents

A patent application on the materials described has been filed with the US and other PTOs.

## Figures and Tables

**Figure 1 biomolecules-13-01559-f001:**
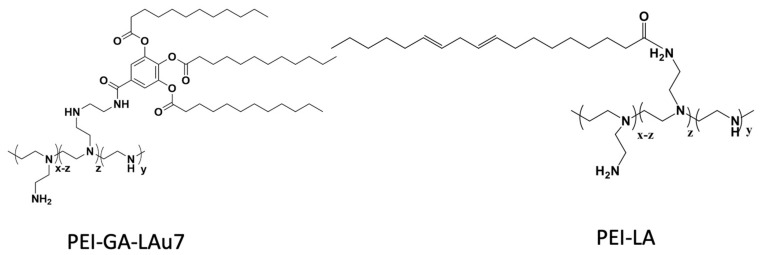
PEI-GA-Lau7 and PEI-LA chemical structures.

**Figure 2 biomolecules-13-01559-f002:**
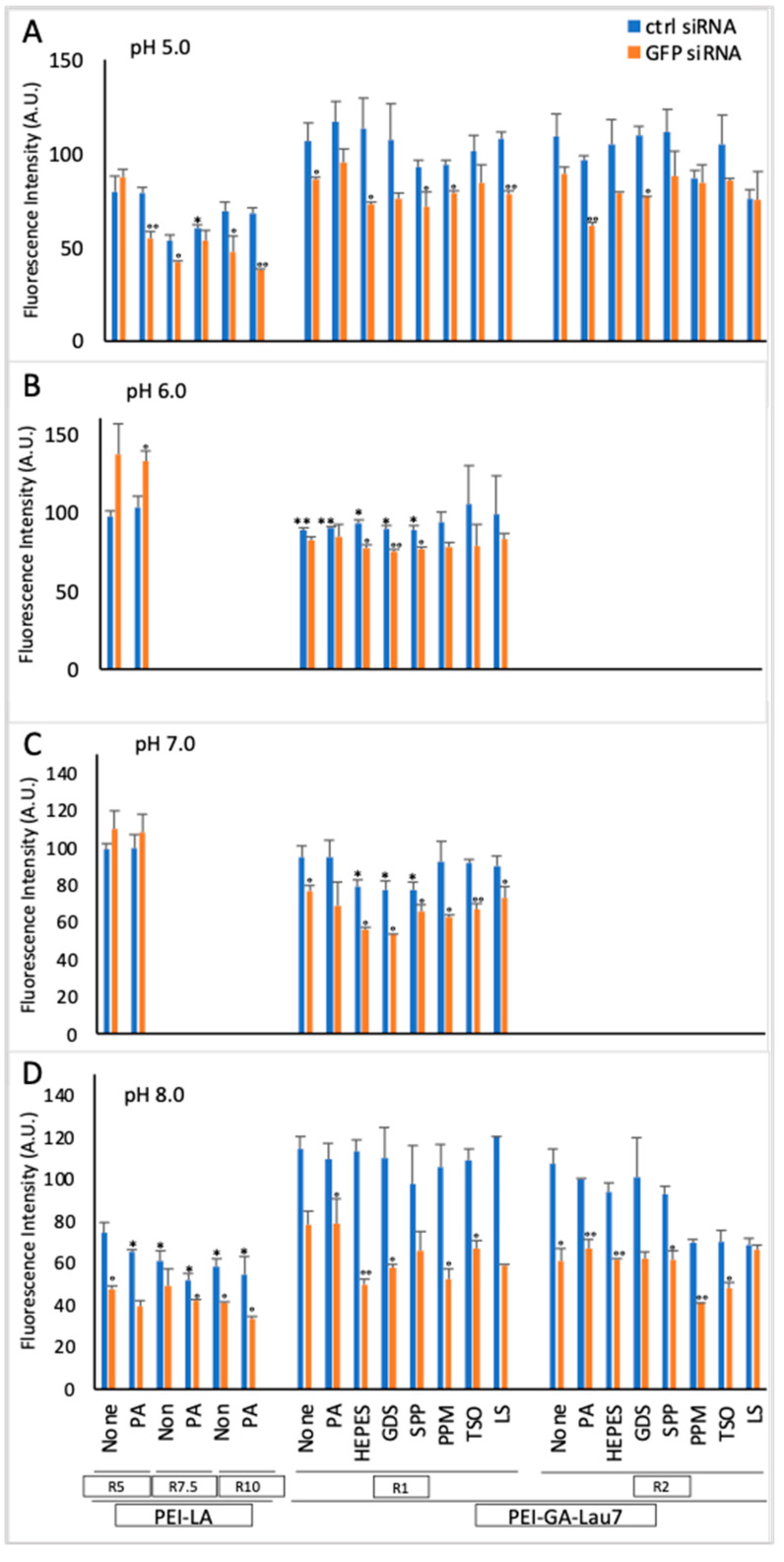
Green fluorescence measurement (microplate reader) in MDA-MB-231 GFP+ cells as an indication of gene expression. The expression of the GFP gene was measured 3 days after delivery of 50 nM siRNA polyplexes at weight ratios 1:1:1 and 2:1:1 of PEI-LA or PEI-GA-Lau7:additive:siRNA which were prepared in phosphate buffer (**A**) pH 5.0, (**B**) pH 6.0, (**C**) pH 7.0 and (**D**) pH 8.0 to MDA-MB-231 cells. NT: No Treatment. The asterisks indicate the significant toxicity of GFP siRNA treatment (*) *p* < 0.05 and (**) *p* < 0.005 compared to no treatment. The circles represent the significant silencing of GFP transcripts by specific siRNA treatment compared to that of control siRNA, (°) *p* < 0.05, (°°) *p* < 0.005.

**Figure 3 biomolecules-13-01559-f003:**
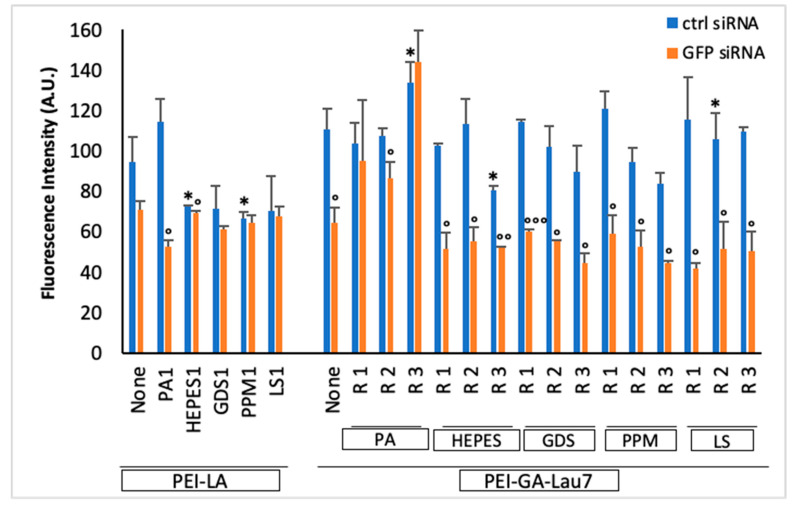
The expression of the GFP gene was measured by a microplate reader. MDA-MB-231 cells were analyzed 3 days after treatment with in-phosphate pH 8.0 siRNA polyplexes. Ratio 1 of PEI-LA or PEI-GA-Lau7 polymer was used in combination with *w*/*w*/*w* ratios 1, 2 or 3 of additives or PA to siRNA for PEI-GA-Lau7 and in the case of PEI-LA, only ratio 1 of additives was used. siRNA was used in a final concentration of 50 nM. NT: No Treatment. The asterisks indicate the significant toxicity of GFP siRNA treatment (*) *p* < 0.05 compared to no treatment. The circles represent the significant silencing of GFP transcripts by specific siRNA treatment compared to that of control siRNA, (°) *p* < 0.05, (°°) *p* < 0.005 and (°°°) *p* < 0.0005.

**Figure 4 biomolecules-13-01559-f004:**
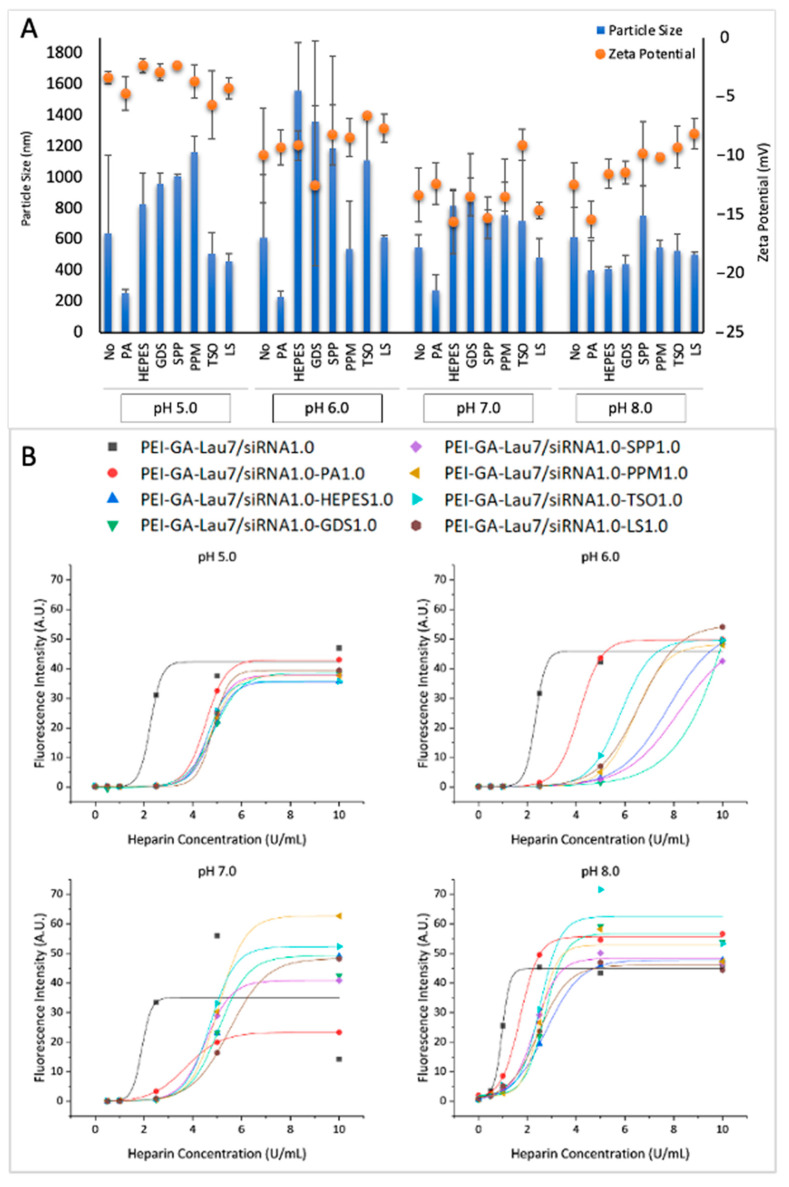
Physiochemical characteristics of additive polyplexes. (**A**) size and 𝜁 potential, and (**B**) stability against heparin for siRNA complexes were measured in different pH levels either with or without additive incorporation at the ratio 1:1:1 of PEI-GA-Lau7:additive:siRNA.

**Figure 5 biomolecules-13-01559-f005:**
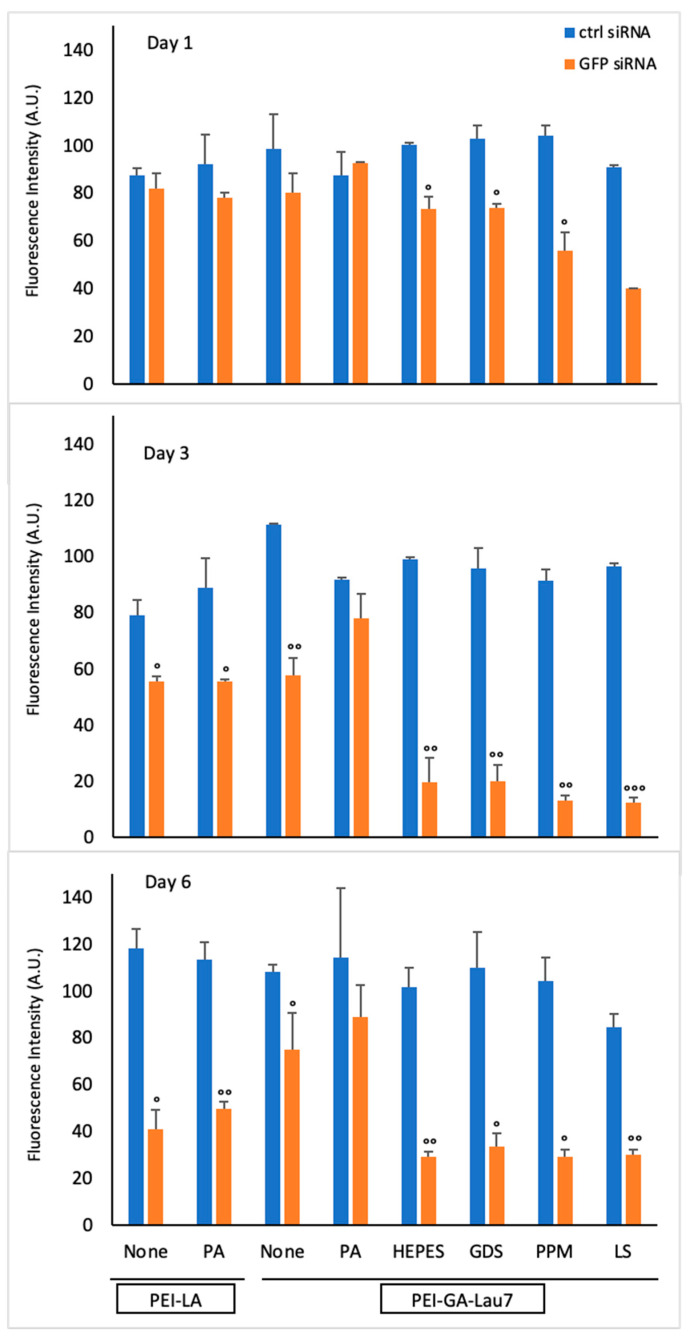
Time course study to reveal the persistence of the GFP silencing effects of pre-optimized siRNA polyplexes. MDA-MB-231 cells were treated with siRNA complexes at the weight ratio 1:1:1 of PEI-LA or PEI-GA-Lau7: additive or PA: siRNA. The amount of green fluorescence was measured by microplate reader 1, 3 or 6 days post-transfection. A total of 50 nM concentration of siRNA was used. The circles represent the significant silencing of GFP transcripts by specific siRNA treatment compared to that of control siRNA, (°) *p* < 0.05, (°°) *p* < 0.005 and (°°°) *p* < 0.0005.

**Figure 6 biomolecules-13-01559-f006:**
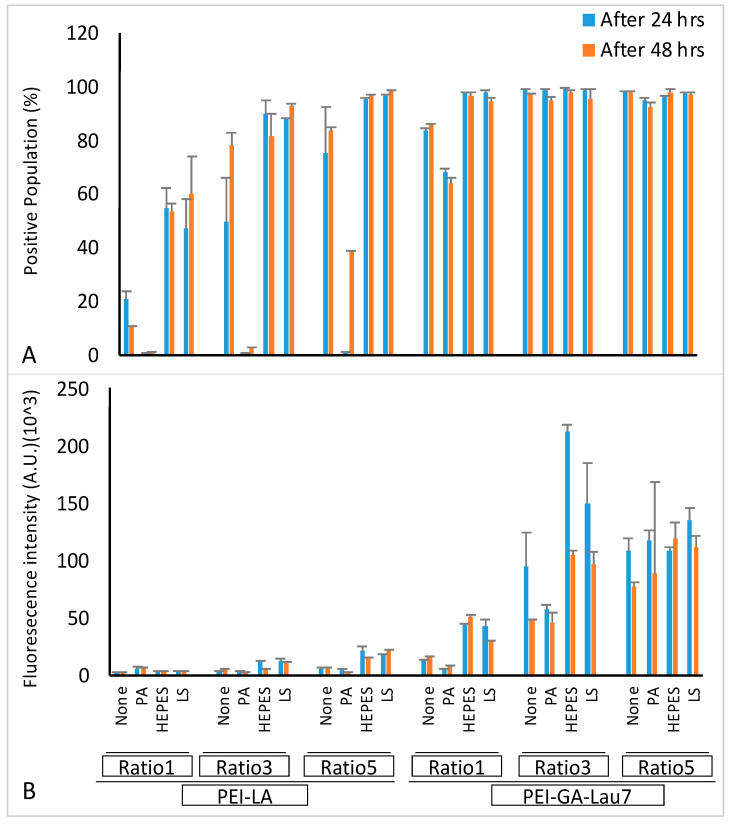
Cellular uptake of siRNA/polymer polyplexes in MDA-MB-231 cells. Cells were transfected with 50 nM of FAM-labeled siRNA at 1:1:1, 1:3:1 or 1:5:1 PEI-GA-Lau7 or PEI-LA:siRNA:additive or PA (*w*/*w*/*w*) ratios either 24 h or 48 h after seeding. The complexes were made in phosphate pH 8.0. (**A**) Mean FAM-labeled siRNA uptake (mean + SD) (**B**) Fam-labeled siRNA positive cells (mean + SD).

**Figure 7 biomolecules-13-01559-f007:**
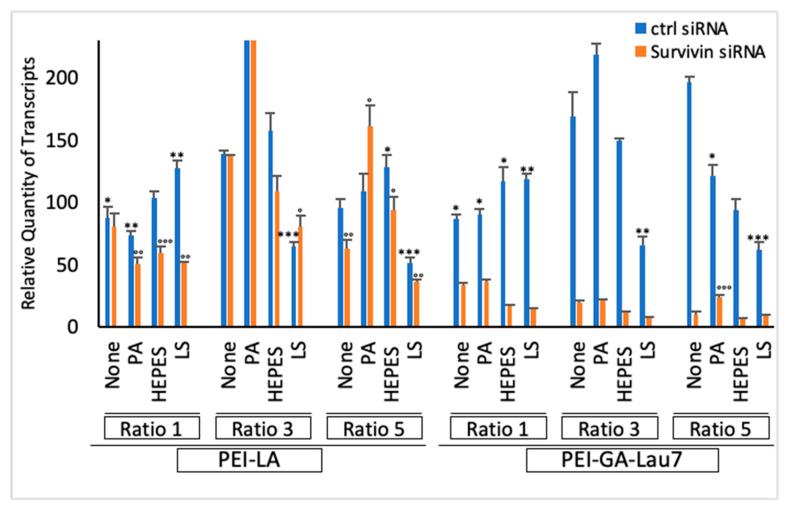
siRNA complexes’ efficiency was examined using qRT-PCR. In total, 300 uL of MDA-MB-231 cells with 20 K per mL confluency were cultured in 48-well plates. After 24 h, 100 uL of survivin siRNA polyplexes were added to the cells. Polyplexes were synthesized in phosphate pH 8.0 and the final concentration of 50 nM of siRNA was used. Ratio 1 of PA, HEPES and LS to siRNA was used as additives. The RNA was extracted 3 days post-treatment. NT: No Treatment. The asterisks indicate the significant toxicity of GFP siRNA treatment (*) *p* < 0.05, (**) *p* < 0.005 and (***) *p* < 0.0005 compared to no treatment. The circles represent the significant silencing of survivin transcripts by specific siRNA treatment compared to that of control siRNA, (°) *p* < 0.05, (°°) *p* <0.005 and (°°°) *p* < 0.0005.

**Figure 8 biomolecules-13-01559-f008:**
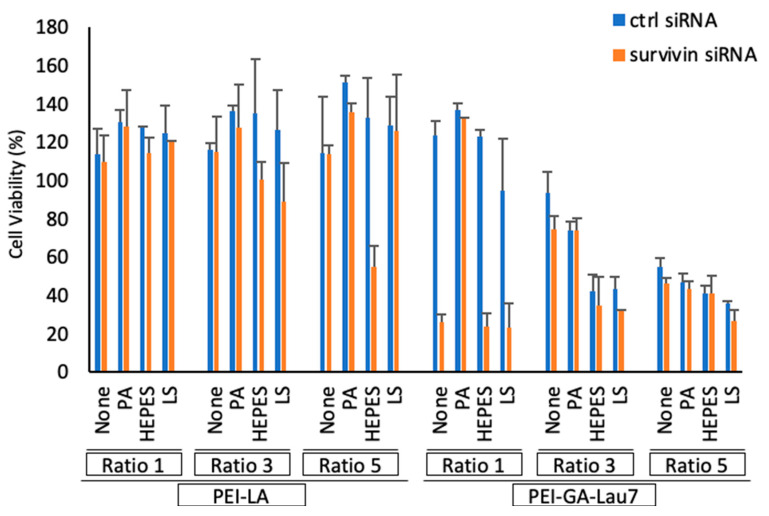
The viability of MDA-MB-231 cells was assessed after the delivery of survivin siRNA in MTT assay. A total of 50 nM siRNA was delivered to the cells 24 h after seeding and the MTT solution (1 mg/mL) was added to the cells 72 h after treatment. Lastly, the absorbance was measured on the same day. Ratios 1, 3 and 5 of PEI-LA and PEI-GA-Lau7 and ratio 1 of additives to siRNA were used. Phosphate pH 8.0 was used as the complexation media. The results are represented as a relative percentage to NT which was put as 100%. NT: No Treatment.

**Table 1 biomolecules-13-01559-t001:** Example of a study design and corresponding siRNA complex preparation (for duplicate wells). Components were added in the order they appear in the chart (left to right), except for the media which was added in two parts (final concentration of 50 nM of siRNA in culture medium). NT: No Treatment which indicates the cells treated with only media as the negative control. Specific (W/W/W) ratios used for polymer/additive/siRNA are mentioned in each experiment separately. The ratio column indicates the ratio of transfection reagent to siRNA.

Carrier	Ratio (*w*/*w*)	Media (uL)	siRNA (uL)	Additive (uL)	Transfection Reagent (uL)	Total Volume (uL)
NT	0	200	0	0	0	200
Buffer + siRNA	0	196	4	0	0	200
PEI-LA/SiRNA5.0	5	193.2	4	0	2.8	200
PEI-LA/SiRNA5.0-PA1.0	5	186.4	4	4	5.6	200
PEI-LA/SiRNA7.5	7.5	191.8	4	0	4.2	200
PEI-LA/SiRNA7.5-PA1.0	7.5	183.6	4	4	8.4	200
PEI-LA/SiRNA10.0	10	190.4	4	0	5.6	200
PEI-LA/SiRNA10.0-PA1.0	10	180.8	4	4	11.2	200
PEI-GA-Lau7/SiRNA1.0	1	194.88	4	0	1.12	200
PEI-GA-Lau7/SiRNA1.0-PA1.0	1	189.76	4	4	2.24	200
PEI-GA-Lau7/SiRNA1.0-HEPES1.0	1	189.76	4	4	2.24	200
PEI-GA-Lau7/SiRNA1.0-GDS1.0	1	189.76	4	4	2.24	200
PEI-GA-Lau7/SiRNA1.0-SPP1.0	1	189.76	4	4	2.24	200
PEI-GA-Lau7/SiRNA1.0-PPM1.0	1	189.76	4	4	2.24	200
PEI-GA-Lau7/SiRNA1.0-TSO1.0	1	189.76	4	4	2.24	200
PEI-GA-Lau7/SiRNA1.0-LS1.0	1	189.76	4	4	2.24	200

## Data Availability

The data supporting the reported results can be obtained from the authors upon request.

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
