# Peer review of "Designing Nanomedicines for Breast Cancer Therapy"

_biomolecules, 2023, doi:10.3390/biom13101559_

Round 1
Reviewer 1 Report
1. Authors should clarify in the introduction which type of NPs will be studied.
2. Authors must use a normal "non-breast cancer" cell to assess the selectivity of the NPs. In introduction, line 100, the authors state that one of the objectives of this work is to minimize nonspecific toxicity. How did the authors evaluate this objective?
3. Authors should state PDI (polidispersity index) for their formulation.
4. Figure 6 caption states: NT: No Treatment. However, there is no NT abbreviation in this Figure.
5. Reestructure the phrases in: line 42 "This necessitates the need for novel modes"; line 78 "Lipidic such as liposomes" (insert word formulation or other after lipidic, as a sugestion). Also, other minor gramatical adjusts must be made,
6. Authors should update the bibliography (and also look for citing errors such as in ref. 14 there is no year of publication), since more than 50 out of 68 references are published before 2018.
Some english errors must be corrected.
Reviewer 2 Report
1. The introduction should include more discussion on the background of different siRNA delivery systems.
2. This study focuses on improving PEI-based delivery systems. However, PEI is not included as a control group.
3. The detailed synthesis and characterization of modified PEI should be provided.
4. The mechanism of improvement of siRNA knockdown should be studied, such as cellular uptake, endosomal escape, and stability.
5. It may be difficult to directly use this system for in vivo study. The physicochemical properties (such as big size and surface) are not suitable for in vivo study. Particularly, surface modification of stealth polymer is important (https://doi.org/10.1016/j.addr.2023.114895).
Round 2
Reviewer 1 Report
All questions were address. I recommend to accept the article in present form.
Author Response
We appreciate your comments and time.
Regards,
Reviewer 2 Report
The authors addressed the concerns to some extent. However, some important information is lacking.
1. For Q2, the authors cited a paper [ DOI: 10.1016/j.actbio.2016.01.025] to say that they previously have already studied PEI-1.2K as a control. However, this paper focused on pDNA. pDNA and siRNA delivery systems are two different stories. Please include PEI as a control group in this study.
2. For Q4, there is no evidence on endosomal escape ability (cell experiments) and siRNA stability (such as degradation). To clarify the mechanism of improvement of siRNA knockdown, these experiments are needed.
3. This paper is very similar to the authors' previous publications, from polymer used, siRNA delivery, and breast cancer (https://doi.org/10.1021/acsabm.2c00978; https://doi.org/10.1021/acs.biomac.8b00918). The authors also mentioned that in Q3. It is difficult to understand why the authors should publish this work.
4. For Q5, it is better to discuss in the paper on stealth polymer (https://doi.org/10.1016/j.addr.2023.114895). It can be the future studies. Stealth polymer is for not only long blood circulation but also good formulation (such as a good example of lipid/mRNA vaccine with PEG used).
